# Self-Selected Pacing during a 24 h Track Cycling World Record

**DOI:** 10.3390/ijerph16162943

**Published:** 2019-08-16

**Authors:** Beat Knechtle, Thomas Rosemann, Pantelis Theodoros Nikolaidis

**Affiliations:** 1Medbase St. Gallen Am Vadianplatz, 9000 St. Gallen, Switzerland; 2Institute of Primary Care, University of Zurich, 8006 Zurich, Switzerland; 3Exercise Physiology Laboratory, 18450 Nikaia, Greece; 4School of Health and Caring Sciences, University of West Attica, 11244 Athens, Greece

**Keywords:** bike, ultraendurance, athlete, cycling speed, power output

## Abstract

The present case study analyzed the pacing in a self-paced world record attempt during a 24 h track cycling event by the current world record holder. The cyclist completed 3767 laps on a 250 m long cycling track and covered a total distance of 941.873 km, breaking the existing world record by 37.99 km. The average cycling speed was 39.2 ± 1.9 km/h (range 35.5–42.8 km/h) and the power output measured was 214.5 ± 23.7 W (range 190.0–266.0 W) during the 24 h of cycling. We found a positive pacing result with negative correlations between cycling speed (*r* = −0.73, *p* < 0.001), power output (*r* = −0.66, *p* < 0.001), and laps per hour (*r* = −0.73, *p* < 0.001) and the covered distance. During the 24 h, we could identify four different phases: the first phase lasting from the start till the fourth hour with a relatively stable speed; the second phase from the fourth till the ninth hour, characterized by the largest decrease in cycling speed; the third phase from the ninth hour till the 22nd hour, showing relatively small changes in cycling speed; and the last phase from the 22nd hour till the end, presenting a final end spurt. The performance in the 24 h track cycling was 45.577 km better than in the 24 h road cycling, where the same athlete cycled slower but with higher power output. In summary, the current world-best ultracyclist covered more kilometers with less power output during the world record 24 h track cycling than during his world record 24 h road cycling. This was most probably due to the more favorable environmental conditions in the velodrome, which has no wind and stable temperatures.

## 1. Introduction

Pacing in sports describes how an athlete distributes work and energy during an endurance performance [1]. To date, six different pacing strategies have been described: all-out, positive, negative, even, parabolic-shaped, and variable pacing strategies [1,2]. Pacing has been explored for use in track cycling [3], road cycling [4], and in ultraendurance cycling [5,6]. Furthermore, the 1 h or 24 h cycling world records [6,7] are a specific discipline, consisting of attempts to break a time-limited (i.e., aim to cover the longest distance in a given time) world record.

It is well known that physiological (training [8], performance level [5,9], gross efficiency [10]) and psychological (motivation [11,12], emotion [13], motivational self-talk [14]) variables influence cycling performance in time trials. For instance, the fastest ultraendurance cyclists have been shown to adopt a more even pacing (i.e., less decrease in speed across race) than their slower counterparts [5]. On the other hand, nonelite cyclists—who seem to be enthusiasts and start a race fast—would be expected to decrease speed largely across a race [11,12]. Furthermore, performance might depend on nutritional (fluid intake [8,15,16], hydration status [17,18], food intake [8,19,20]) and other external/technical (prize money [21], time of day [22], equipment [8], starting strategy [3,23], and pacing strategy [3,12,24,25,26]) variables. Overall, the influence of environmental conditions, such as heat, may have the highest impact on cycling performance [11,27,28,29,30,31], with performance generally being impaired when cycling in hot environment [32,33,34].

In time trial cycling, different kinds of time- or distance–limited (i.e., aim to cover a given distance in the shortest time) trials have been investigated [35]. For time-limited cycling, a very specific discipline is the 24 h time trial cycling event [36], where very little is known about the pacing strategy of the athletes. Recently, a case study investigated the pacing in a self-paced world record attempt in 24 h road cycling [6]. The authors found that cycling speed gradually declined during the 24 h (i.e., positive pacing), and environmental influences, such as temperature and wind, showed an effect on performance, with lower temperature and stronger wind being associated with faster race speed. Although this case study improved our understanding of pacing in ultraendurance cycling, its findings could not be generalized to track cycling, which has unique environmental characteristics [37]. Furthermore, limited information exists regarding pacing strategies in track cycling, and the available information concerns relatively short distances [38,39].

Therefore, the aim of the present case study was to examine the attempt of a cyclist to break the current world record in 24 h track cycling. Furthermore, we compared the pacing strategies between the outdoor and indoor trials. We hypothesized that the athlete would achieve longer distance at a given time (24 h) in the indoor trial as he would have minimal environmental influences compared to the outdoor trial, with factors such as temperature, humidity, and air pressure all being typically under thermostatic control indoors.

## 2. Materials and Methods

### 2.1. Ethics Approval

All procedures used in the study were approved by the Institutional Review Board of Ethikkommission St Gallen, Switzerland, with a waiver of the requirement for informed consent of the participant given the fact that the study involved the analysis of publicly available data (EKSG 01-06-2010). Written informed consent was obtained by email from the athlete, and he agreed to the analysis and publication of his data as presented in this article.

### 2.2. The Cyclist

The cyclist (35 years, 76 kg, 1.86 m, body mass index (BMI) 22 kg/m^2^) is an experienced ultraendurance cyclist. To date, he has won several long-distance cycling races. For instance, he holds the record for the fastest Crossing of America in the “Race across America” (RAAM) in 2014 in 7:15:56 d:h:min with an average cycling speed of 26.43 km/h. RAAM crosses the United States from west to east (4860 km and 35,000 m of altitude). In 2015, he set a world record in the 24 h road cycling event, where he cycled 896 km at an average speed of 37.33 km/h [40].

### 2.3. The Event

At noon on October 14, 2017, the cyclist started his record attempt in the Tissot Velodrome [41] in Grenchen, Switzerland. The track is a 250 m long and 7 m wide oval built out of wood. Temperature in the velodrome was kept constant at ~20 °C, and humidity was low at ~30% throughout the whole event. The previous record was set on October 8–9, 2010, by Marko Baloh (Slovenia) at the Montichiari Velodrome in Brescia, Italy, by completing 3615 full laps of 250 m and covering a full distance of 903.76 km [42].

The cyclist used a conventional time trial bike (Shiv, S-WORKS, Specialized) with a disc wheel. He used Roval wheels [43] with specially made Turbo Cotton tires. The front brake was omitted for the purpose of aerodynamics; a brake lever and one chain ring had to suffice. The chainwheel had 53 sprockets, while the gear rim pinion had 11–23 sprockets. He changed the gears only to change the pedaling frequency. Power output in W was measured using Power2max NG (Saxonar GmbH, Waldhufen, Germany) [44], which the participant had also used in his last record race. After each hour, the accumulated distance was recorded. The athlete tried to maintain the power output of 230–260 W. Lap times were recorded electronically.

### 2.4. Statistical Analysis

The acceptable type I error was set at *p* < 0.05. All data are presented as means and standard deviations. The data were checked for normality using visual inspection of probability–probability plots and the Kolmogorov–Smirnov test. Figures were created using GraphPad Prism v. 7.0 (GraphPad Software, San Diego, CA, USA); all other statistical analyses (descriptive statistics and correlation analysis) were carried out using IBM SPSS v. 23.0 (SPSS, Chicago, IL, USA). The relationship of cycling speed, power, and laps per hour with covered distance, as well as that between cycling speed and power, during the race was examined using the Pearson moment correlation coefficient *r*. In addition, the coefficient of determination (*R*^2^) was calculated. The magnitude of these relationships was evaluated as trivial (*r* < 0.10), small (0.10 ≤ *r* < 0.30), moderate (0.30 ≤ *r* < 0.50), large (0.50 ≤ *r* < 0.70), very large (0.70 ≤ *r* < 0.90), and almost perfect (*r* ≥ 0.90) [45]. Data from the record attempt in 24 h track cycling are presented together with the data from the record attempt in 24 h road cycling from 2015 [6]. Standardized differences in means (Cohen’s *d*) between indoor and outdoor races were calculated for each phase of the two races.

## 3. Results

At the beginning of the attempt, the athlete cycled at an average speed of 41 km/h. His support team provided him food and drinks, mainly consisting of water, isotonic sports drinks, and green tea. They provided 400–500 kcal of energy per hour. After 2–3 h, the cyclist suffered from digestive problems, most likely due to the position on the time trial bike. To solve the problem of digestion, the support crew gave him more green tea plus Coca Cola^®^ and Ensure^®^. After the first two hours, he already had his first problems when low back pain occurred. Moreover, an incident of serious emotional distress after two hours of cycling was recorded by his support team. During the night, he had problems with drowsiness, and the support crew gave him coffee and noncaffeinated chewing gum. He made his first stop of 2.5 min for a toilet break after 21:53 h:min of continuous cycling. Up to this point, he had consumed 15 L of fluids. After 23 h of cycling, he broke the existing world record. In the last hour, he tried to increase his cycling speed [46].

The cyclist covered 941.873 km and completed 3767 laps in 24 h. After 23 h and 10 min, he broke the existing world record. The cycling speed was 39.2 ± 1.9 km/h (range 35.5–42.8 km/h), and the power output was 214.5 ± 23.7 W (range 190.0–266.0 W). The number of laps per hour was 157.0 ± 7.5 (range 142–171), and the time per lap was 22.98 ± 1.07 s (21.05–25.35 s).

Figure 1, Figure 2 and Figure 3 show the relationship of speed, power, and number of laps, respectively, with the distance. In all cases, a negative correlation was observed, ranging from large (i.e., power output) to very large (i.e., cycling speed and number of laps) magnitude. These correlations indicated a positive pacing strategy, i.e., a decrease in cycling speed across the event. Furthermore, four phases could be identified using visual inspection of the nonlinear regressions in the figures of speed, power, and number of laps completed per hour during the race: the first phase lasting from the start till the fourth hour with a relatively stable cycling speed; the second phase from the fourth till the ninth hour, characterized by the largest decrease in cycling speed; the third phase from the ninth hour till the 22nd hour, showing relatively small changes in cycling speed; and the last phase from the 22nd hour (approximately after a 2.5 min break for toilet) till the end, presenting a final end spurt. In addition, cycling speed correlated significantly with power output (Figure 4).

When we compared cycling speed (Figure 1) and power output (Figure 2) between the two attempts, we could see that cycling speed was higher in track cycling compared to road cycling, whereas the opposite trend was shown for power output (Table 1). Considering cycling speed (Figure 1), power output (Figure 2), and the correlation between the two (Figure 4), the goodness of fit was stronger for track cycling, highlighting the influence of “other” factors (e.g., temperature, wind) of road cycling. It should be highlighted that the relationship between power output and cycling speed was linear outdoors and curvilinear indoors.

## 4. Discussion

In this case report, we analyzed self-selected pacing during a 24 h track cycling world record. As hypothesized, the athlete’s new world record in 24 h track cycling was better than his world record in 24 h road cycling although his power output was lower. When we compared the two 24 h cycling world records, he had improved on the existing world record in track cycling (903.76 km) held by Marko Baloh by ~38 km (4.2%); in 2015, he had improved on the existing world record in 24 h road cycling of 834.77 km, set in 2004 by the Slovenian Jure Robič, by ~61 km (7.4%).

The performance in the 24 h track cycling was 45.577 km better than in the 24 h road cycling. For the world record attempt in 24 h road cycling, he was riding at a mean cycling speed of 37.34 km/h and achieved an average power output of 250.2 W [6], while the mean cycling speed in the 24 h track cycling world record was 39.2 ± 1.9 km/h with a mean power output of 214.5 ± 23.7 W. It should be highlighted that the cycling speed did not correlate perfectly with the power output. The absence of a perfect relationship between cycling speed and power output should be attributed to the fact that power output depends not only on cycling speed but also on the bicycle’s design, the cyclist’s size and position, and environmental factors, such as wind and road surface [37,47].

In the corresponding world record in road cycling, the decrease in cycling speed and power output throughout the laps could be modeled linearly, and the ambient air temperature and wind speed were related to cycling speed for the whole event. In that event, the air temperature was 13 °C at the start and dropped to 2 °C in the night at 03:00 a.m.; a low temperature is associated with fast speed and vice versa [6]. It is well known that environmental influences, such as heat [31] and cold [32], have an influence on pacing in cycling time trials, with hot temperatures inducing larger decrease in performance, i.e., a more positive pacing. However, heat and cold seem to influence performance differently [32]. A review of the influence of environmental factors (e.g., temperature and wind) on cycling performance concluded that ambient temperature and wind exerted a small effect in outdoor events, whereas power output seemed to be maintained at moderate temperatures during indoor trials [34].

For cyclists competing in the RAAM, the change in temperature and altitude had an influence on cycling speed and power output, with temperature having a positive and altitude having a negative influence on power output for all finishers [5]. In the actual track cycling event, the athlete had no problems with temperature (constant at 20 °C) or wind (no wind), remained in the same sitting position throughout the 24 h, and was therefore able to ride faster (1.86 km/h faster) with a lower power output (35.7 W less on average). The higher cycling speed at a lower power output was most probably due to the surface of the velodrome, which is smoother than the asphalt of a road. A smoother surface decreases rolling resistance compared to asphalt, and a cyclist can consequently attain faster speed for less power output. Moreover, a cyclist can adopt a more aerodynamic position indoors than outdoors with a lower cross-sectional area in a thermostatically controlled environment, reducing wind resistance against a constant air pressure and thus increasing speed [47,48].

Regarding pacing in cycling, differences regarding pacing strategy seem to exist between distance- and duration-based trials of short and long duration. Cyclists starting in distance-based time trials begin the event at relatively higher power outputs than a similar time-based trial. This may result from discrete differences in cyclists’ ability to judge or predict an exercise endpoint when performing time- and distance-based trials [35].

Overall, in our study, the cycling speed decreased continuously during the athlete’s record attempt, corresponding to a positive pacing accompanied by an end spurt. This pacing is the only possibility for ultraendurance cyclists, as has already been reported by the world record in 24 h road cycling [6] and by elite cyclists in the RAAM [5]. The decrease in cycling speed and power output seemed to be curvilinear, as has already been described in this world record attempt in the 24 h road cycling event [6]. Moreover, a positive pacing strategy was observed in mountain bike racing [49] and in a 20 min uphill cycling time trial [50].

However, regarding new studies, an even-pacing strategy would be more appropriate [51] in order to attenuate perturbations in the physiological response and to lower perception of effort in comparison to self- and variable-paced strategies [26]. An even distribution of power output is both physiologically and biophysically optimal for longer time trials held in conditions of unvarying wind and gradient. From a biophysical standpoint, the optimum pacing strategy for road time trials may involve increasing power in headwinds and uphill sections and decreasing power in tailwinds and when travelling downhill [52].

Using visual inspection of the nonlinear regressions in the figures of speed, power, and number of laps completed per hour during the race, four distinct phases during the record attempt were highlighted: (a) a stable speed in the first four hours, (b) a decrease in cycling speed between the fourth and the ninth hour, (c) small changes in cycling speed from the ninth to the 22nd hour, and (d) a final end spurt in the last two hours. This change in cycling speed is most likely explained by aspects such as mood, reduced fuel availability, fatigue, and pain perception, which might have an influence on cycling speed [53,54]. It has been reported that hydration status affects mood in ultracycling, with dehydrated athletes experiencing greater fatigue and pain than euhydrated athletes in an ultracycling race [55].

During the night, the cyclist had problems with drowsiness and suffered from monotonicity. He even reported serious emotional distress after two hours of cycling. The aspect of emotion should not be ignored in ultraendurance performance [56]. It has been shown that cyclists competing in the RAAM reported experiencing an optimal emotional state for less than 50% of the total race time [57]. Furthermore, it has been observed that perception of pain, freshness, and motivation differed between a completed and a noncompleted 100 mile running race [56]. Also, the aspect of caffeinated drinks in cycling performance is of importance. Caffeine use in cycling time trial performance has reported mixed effects, with some studies showing improved performance [58,59,60], whereas other research reported no improvement [61,62]. Such variation on the effectiveness of caffeinated drinks might depend on factors such as heat, familiarization with caffeine consumption, quantity of caffeine consumption, and exercise duration [58,59,60,61,62].

A consideration of the present case study was the specific characteristics of the race in terms of exercise duration and environmental conditions; thus, caution is needed to transfer the findings to other settings. For instance, with regard to the role of environmental conditions, it has been observed that cycling a 30 km time trial was faster in 24 °C than in 35 °C (68% relative humidity in both cases) [27]. Furthermore, as a field study that relied on secondary analysis of data produced by the cyclist, it presents the inherent pros (e.g., close to the reality of the sport) and cons (e.g., limited control over variables compared to a laboratory setting). The strength of this case study is that the same participant performed attempts in both road [6] and track cycling (present study), and this offered the opportunity to compare the two cycling formats during a similar time period (24 h). Furthermore, the cyclist used the same bike and the same system to measure power output in both cases (road and track cycling). It has previously been shown that technical and equipment characteristics, e.g., riding with hands on the brake woods and using an aerodynamic wheel set, would improve performance [8]. Thus, the findings of the present study are of great practical value for cycling coaches as well as scientists interested in the differences between these two formats of 24 h cycling.

## 5. Conclusions

In the present world record in the 24 h track cycling event, the current world-best ultracyclist achieved more kilometers with less power output than in his world record in the 24 h road cycling competition [6]. Considering the more favorable environmental conditions on the track (i.e., no wind and stable temperatures) than in the road cycling world record, it is reasonable to assume that these environmental conditions influenced pacing and power output during the 24 h world record attempt in ultracycling.

## Figures and Tables

**Figure 1 ijerph-16-02943-f001:**
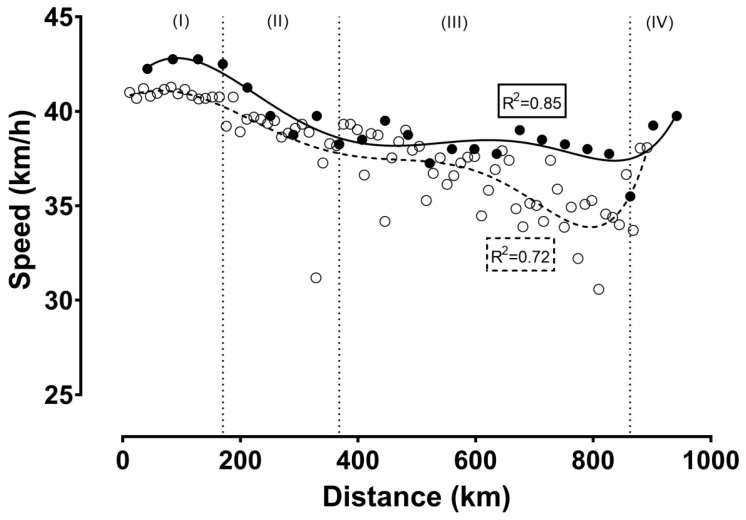
Cycling speed (km/h) during the record attempt. Latin numbers (I, II, III, IV) denote the phases of the race, which are separated by vertical dashed lines. The present record attempt (track cycling) is presented by ●, whereas the previous attempt (road cycling) is presented by ◯. *R*^2^ estimates the goodness of fit.

**Figure 2 ijerph-16-02943-f002:**
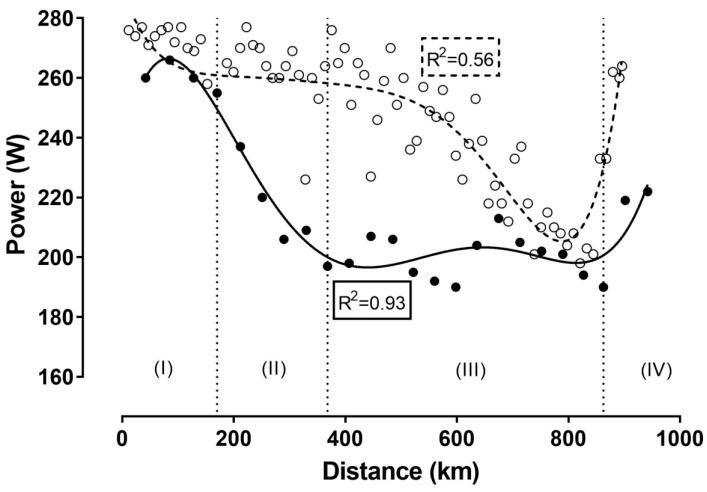
Power output (W) during the record attempt. Latin numbers (I, II, III, IV) denote the phases of the race, which are separated by vertical dashed lines. The present record attempt (track cycling) is presented by ●, whereas the previous attempt (road cycling) is represented by ◯. *R*^2^ estimates the goodness of fit.

**Figure 3 ijerph-16-02943-f003:**
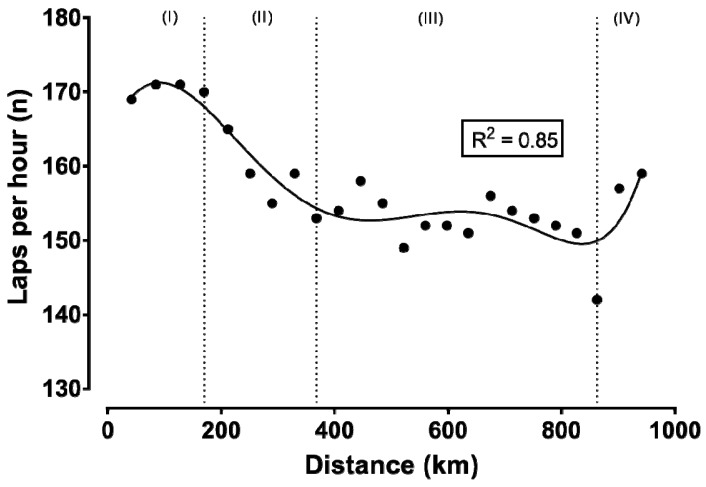
Number of laps per hour during the record attempt. Latin numbers (I, II, III, IV) denote the phases of the race, which are separated by vertical dashed lines. The previous attempt (road cycling) is not presented due to different lap distance. *R*^2^ estimates the goodness of fit.

**Figure 4 ijerph-16-02943-f004:**
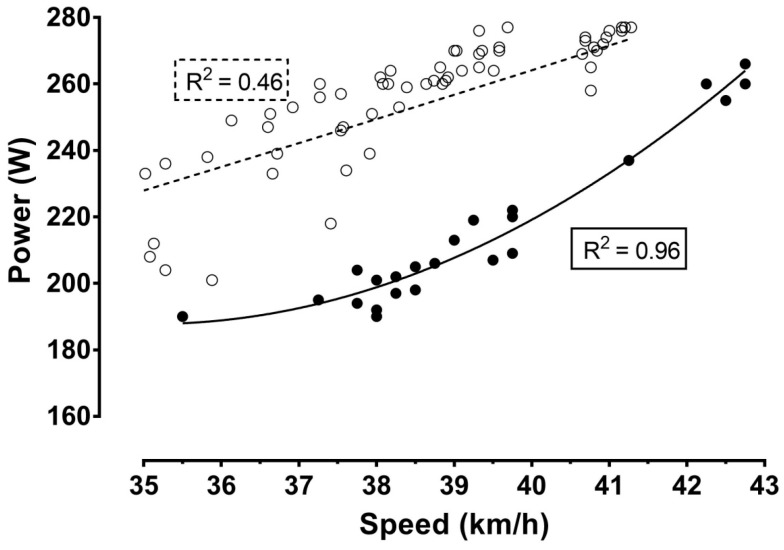
Relationship between power output and cycling speed. Latin numbers (I, II, III, IV) denote the phases of the race, which are separated by vertical dashed lines. The present record attempt (track cycling) is presented by ●, whereas the previous attempt (road cycling) is represented by ◯. *R*^2^ estimates the goodness of fit.

**Table 1 ijerph-16-02943-t001:** Comparison between indoor and outdoor performances by phase of the race. d = (Cohen’s *d*)

Phase	Speed (km/h)			Power (W)		
	Indoor	Outdoor	Difference	Outdoor	Indoor	Difference
I	42.56 ± 0.24	40.93 ± 0.21	d = 7.23	260.3 ± 4.5	274.9 ± 2.4	d = 4.05
II	39.55 ± 1.15	38.76 ± 2.12	d = 0.46	213.8 ± 15.4	263.8 ± 12.4	d = 3.58
III	38.06 ± 0.97	36.36 ± 2.02	d = 1.07	199.8 ± 7.2	238.8 ± 22.8	d = 2.31
IV	39.50 ± 0.35	35.81 ± 2.03	d = 2.53	220.5 ± 2.1	236.6 ± 27.0	d = 0.84

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
