# Peer review of "Self-Selected Pacing during a 24 h Track Cycling World Record"

_ijerph, 2019, doi:10.3390/ijerph16162943_

Round 1

Reviewer 1 Report

Congratulation to the authors for the nice work conducted. I would recommend some change to improve manuscript quality

Introduction

Try to explain physiological bases of different pace strategies, as well as, how psychological characteristic of athletes could influence the election of each one

Discussion

Discuss the effect of environmental conditions in the physiological response of athletes and how its could affect pace and race time

Author Response

Comments and Suggestions for Authors

Congratulation to the authors for the nice work conducted. I would recommend some change to improve manuscript quality

Answer: We thank the expert reviewer for the constructive comments. Please, find our detailed answers to the comments below and the changes within the text highlighted in red.

Introduction

Try to explain physiological bases of different pace strategies, as well as, how psychological characteristic of athletes could influence the election of each one

Answer: We agree with the expert reviewer and added this aspect in the introduction („For instance, the fastest ultra-endurance cyclists have been shown to adopt a more even pacing (i.e. less decrease of speed across race) than their slower counterparts [5]. On the other hand, non-elite cyclists - who seem enthusiasts and start fast a race - would be expected to decrease speed largely across a race [11,12].“).

Discussion

Discuss the effect of environmental conditions in the physiological response of athletes and how it could affect pace and race time

Answer: We agree with the expert reviewer and enhanced this aspect in the third paragraph of the discussion.

Reviewer 2 Report

The authors present a case study of a successful 24-hour indoor cycling world record attempt. The manuscript is generally well written, with only moderate English language changes recommended and some further clarification required at points of the manuscript. Further detail is required around statistics, and there are more comparisons to be made here that I feel would strengthen the paper. Some further explanation of points raised in the discussion is also warranted. I feel the paper is novel and of interest to readers, but is also a complement to the authors' previous body of work together. I present line by line recommendations below:

Abstract

Introduction

Line 32: remove 'such as' and replace with a colon - such as would suggest you're providing an example or two, not a complete list as you've provided

Line 33: remove 'In cycling'

Line 34-36: consider rearranging this sentence to read: Further, the 1-h or 24-h cycling world records [6,7] are a specific discipline, consisting of attempts to break a time-limited world record.

Line 37-41: this sentence is really long. Whilst it is well referenced, it may be more appropriate to group variables by physiological, psychological and external/technical factors and write a short sentence for each, or simply reference under these broader subject headings as these are not the focus of the present paper.

Page 2, Line 1: Are the terms time- or distance-limited trials used by other authors?

Line 6: 'showed an effect on performance' - was this positive or negative? 

Line 12: please amend to 'pacing strategies'

Line 13: please state distance as opposed to km

Line 14: I would disagree that the environment has no influence upon performance in this trial. I would recommend changing 'no' to predictable, minimal or consistent as factors such as temperature, humidity and thus air pressure are all typically under thermostatic control within a velodrome so still have an effect but less so than an outdoor trial.

Materials and Methods

Line 18: Please include the institutional review board

Line 23-27: Please break this sentence up into something more manageable for the reader.

Line 31: consider changing 'broad' to wide

Line 36: can you include details of gearing and braking here please? I presume this was a fixed gear, but what size gear would surely be of interest to readers

Lines 37-38: the sentence that starts 'In contrast...' is perhaps unnecessary as specific contrasts to the other record aren't stated

Line 41: 'split value' should read 'distance split' or 'accumulated distance'

Line 46 onwards: comments regarding digestion/gastrointestinal upset, back pain etc. are better suited to results as they are an effect of the exercise bout. I'd recommend including a section on athlete perceptions or athlete feedback to allow for these comments to be appropriately documented, as they are of genuine value and add to the usefulness of the manuscript

Page 3 Statistical analysis - did you check normality of the data in order to use a Pearson correlation coefficient? If data are non-normally distributed the non-parametric equivalent Spearman's correlation may be more appropriate. Also, your figures show Rvalues, but there is no mention of this statistic in the statistical analyses section. A brief comment regarding goodness of fit or variance explained would suffice. As you present so many Rvalues is it worth running a regression analysis? Also, I would recommend comparing the indoor to the outdoor data in some way e.g. an effect size and confidence interval.

Line 12-13: please use the term 'large' as opposed to 'great' as per Hopkins descriptors; the same applies for the results

Results

Line 28/29: why did you decide the partition data from hour 22 when you note an increase in speed from hour 23? Was this to align with the toilet break?

Is there a way to document the phases of the indoor attempt on Figure 4? Perhaps by using differently shaped symbols? Only do this if you feel it adds to your subsequent discussion. It is also interesting to note that the relationship between power and speed in indoor and outdoor cycling are differently shaped. If you were to plot outdoor as a curve (like indoor values) I suspect you would have a different R2 value for outdoor than you currently do.

Discussion

Line 25/26: environmental factors need to be mentioned in this sentence too.

Page 6 Line 5: please provide the 'so what' here. A smoother surface presents less rolling resistance than asphalt meaning the rider can attain higher speeds, for less power output. This needs to be stated. The rider may also have been able to adopt a more aerodynamic position indoors than outdoors, again a lower cross sectional area in a thermostatistically controlled environment reduces wind resistance in a consistent air pressure, also facilitating higher cycling speed.

Line 6-10: I'm unsure what this adds by being placed here, it may fit one paragraph earlier, or may be superfluous.

Line 14-15: you suggest a linear relationship to power/speed reduction but figure 4 shows a curve for the present dataset?

Line 16-19: there is a lot of information here, I'm unsure of all of its relevance to the present study unless it is written more directly and links to the present investigation e.g. Similarly, a non-linear decrease in power-output and cycling speed was seen in finishers of the RAAM, with a sharper decrease in cycling speed and power-output seen in the closing stages than the previous 20 checkpoints [5]. Similar pacing strategies have also been observed in mountain bike racing [47] and a 20-min uphill cycling time trial [48].

Line 31-32: I don't think you can make this comment as you've not measured/reported energy intake, or gastrointestinal symptoms in any quantifiable way. The athlete was also fed a variety of drinks throughout the bout, so was this combination or a single drink the cause of the GI distress? I'd suggest removing this comment unless you have ample nutrition data that you're also willing to present in the present paper.

Line 37: What does monotonicity mean in this context? Is there an alternative definition?

Line 38: 'serious emotional distress' seems like it would cause cessation of exercise. Was this managed, measured or self-reported in some way? If so this needs to be explained in the methods and results. Recommend the ultra-running case study by Best et al., (2018) in Sports as a reference that assessed emotional factors over a 24hour duration, as opposed to the RAAM. 

Line 41-44: these sentences are in direct contradiction to each other. I suggest combining to 'caffeine use in cycling time trial performance has reported mixed effects [55-59], depending upon factors such as...'. If this is to be included, greater specificity is required.

Lines 45-49. This paragraph needs to come earlier. You raise important points that would lead into your discussion around pacing strategies between cycling durations and disciplines.

I don't necessarily think you have a set of limitations as you did not design this experiment, it is a secondary analysis of the data produced by the athlete during the attempt. I would reframe this section as 'considerations'. This applies to things such as bike frame, testing surface, wheel choice, environment, nutritional strategies etc.

Conclusions

I think your conclusion emphasises the need for a direct comparison between trials as opposed to comparing correlation magnitudes and R2 values. A simple standardised difference in means at each phase of each attempt would suffice, this way you can attribute differences in global performance between attempts not just to environmental factors, but to performance in specific sections of the race.

Author Response

Comments and Suggestions for Authors

The authors present a case study of a successful 24-hour indoor cycling world record attempt. The manuscript is generally well written, with only moderate English language changes recommended and some further clarification required at points of the manuscript. Further detail is required around statistics, and there are more comparisons to be made here that I feel would strengthen the paper. Some further explanation of points raised in the discussion is also warranted. I feel the paper is novel and of interest to readers, but is also a complement to the authors' previous body of work together. I present line by line recommendations below:

Answer: We thank the expert reviewer for the detailed comments. Please, find our point-by-point answers to the specific comments below. The requested changes are highlighted in red within the text.

Abstract

Introduction

Line 32: remove 'such as' and replace with a colon - such as would suggest you're providing an example or two, not a complete list as you've provided

Answer: We agree with the expert reviewer and corrected it as suggested.

Line 33: remove 'In cycling'

Answer: We agree with the expert reviewer and corrected it as suggested.

Line 34-36: consider rearranging this sentence to read: Further, the 1-h or 24-h cycling world records [6,7] are a specific discipline, consisting of attempts to break a time-limited world record.

Answer: We agree with the expert reviewer and corrected it as suggested.

Line 37-41: this sentence is really long. Whilst it is well referenced, it may be more appropriate to group variables by physiological, psychological and external/technical factors and write a short sentence for each, or simply reference under these broader subject headings as these are not the focus of the present paper.

Answer: We agree with the expert reviewer and splited into shorted sentences as suggested („It is well known that physiological (training [8], performance level [5,9], gross efficiency [10]) and psychological variables (motivation [11,12], emotion [13], motivational self-talk [14]) influence cycling performance in time trials. Furthermore, performance might be depend on nutritional (fluid intake [8,15,16], hydration status [17,18], food intake [8,19,20]) and other external/technical variables (prize money [21], time of day [22], equipment [8], starting strategy [3,23] and pacing strategy [3,12,24-26]).“.

Page 2, Line 1: Are the terms time- or distance-limited trials used by other authors?

Answer: We agree with the expert reviewer that they are not widely used in the literature, and thus, we defined in the first time appeared in the text (for time-limited, see „(i.e. aim to cover the longest distance at a given time)“ in the end of the first paragraph and for distance-limited, see  „(i.e. aim to cover a given distance at the shortest time)“ in the beginning of the third paragraph of the introduction.

Line 6: 'showed an effect on performance' - was this positive or negative? 

Answer: We agree with the expert reviewer and clarified it („where a lower temperature and a stronger wind were associated with faster race speed“).

Line 12: please amend to 'pacing strategies'

Answer: We agree with the expert reviewer and corrected it accordingly.

Line 13: please state distance as opposed to km

Answer: We agree with the expert reviewer and corrected it to „would achieve longer distance at a given time (24 hours)“.

Line 14: I would disagree that the environment has no influence upon performance in this trial. I would recommend changing 'no' to predictable, minimal or consistent as factors such as temperature, humidity and thus air pressure are all typically under thermostatic control within a velodrome so still have an effect but less so than an outdoor trial.

 Answer: We agree with the expert reviewer and revised this sentence to „minimal environmental influences compared to the outdoor trial, since factors such as temperature, humidity and air pressure would be all typically under thermostatic control indoors.“.

Materials and Methods

Line 18: Please include the institutional review board

Answer: We agree with the expert reviewer and added it (St Gallen, Switzerland).

Line 23-27: Please break this sentence up into something more manageable for the reader.

Answer: We agree with the expert reviewer and broke it into shorter sentences.

Line 31: consider changing 'broad' to wide

Answer: We agree with the expert reviewer and corrected it accordingly.

Line 36: can you include details of gearing and braking here please? I presume this was a fixed gear, but what size gear would surely be of interest to readers

Answer: We agree with the expert reviewer and added ‘The chainwheel had 53 sprockets, the gear rim pinion from 11 to 23 sprockets. He changed the gears only to change the pedalling frequency’, see https://bikeboard.at/Board/24-h-Bahn-Weltrekord-Christoph-Strasser-th237283

Lines 37-38: the sentence that starts 'In contrast...' is perhaps unnecessary as specific contrasts to the other record aren't stated

Answer: We agree with the expert reviewer and deleted it.

Line 41: 'split value' should read 'distance split' or 'accumulated distance'

Answer: We agree with the expert reviewer and changed it to „the accumulated distance“.

Line 46 onwards: comments regarding digestion/gastrointestinal upset, back pain etc. are better suited to results as they are an effect of the exercise bout. I'd recommend including a section on athlete perceptions or athlete feedback to allow for these comments to be appropriately documented, as they are of genuine value and add to the usefulness of the manuscript

Answer: We agree with the expert reviewer and transferred this part in the beginning of the results in a separate paragraph.

Page 3 Statistical analysis - did you check normality of the data in order to use a Pearson correlation coefficient? If data are non-normally distributed the non-parametric equivalent Spearman's correlation may be more appropriate. Also, your figures show Rvalues, but there is no mention of this statistic in the statistical analyses section. A brief comment regarding goodness of fit or variance explained would suffice. As you present so many Rvalues is it worth running a regression analysis? Also, I would recommend comparing the indoor to the outdoor data in some way e.g. an effect size and confidence interval.

Answer: We agree with the expert reviewer and clarified these aspects. For normality, we added „The data were checked for normality using visual inspection of probability-probability plots and Kolmogorov-Smirnov test.“ in the Methods. Also, we added information about the use of R2 in the Methods and about the new analysis of differences between indoor and outdoor data.

Line 12-13: please use the term 'large' as opposed to 'great' as per Hopkins descriptors; the same applies for the results

Answer: We agree with the expert reviewer and corrected both in the methods and results as suggested.

Results

Line 28/29: why did you decide the partition data from hour 22 when you note an increase in speed from hour 23? Was this to align with the toilet break?

Answer: We agree with the expert reviewer that needs clarification and added the explanation („(approximately after a 2.5 min break for toilet)“).

Is there a way to document the phases of the indoor attempt on Figure 4? Perhaps by using differently shaped symbols? Only do this if you feel it adds to your subsequent discussion. It is also interesting to note that the relationship between power and speed in indoor and outdoor cycling are differently shaped. If you were to plot outdoor as a curve (like indoor values) I suspect you would have a different R2 value for outdoor than you currently do.

Answer: We agree with the expert reviewer and added the aspect of the variation of speed/power relationship depending on indoors or outdoors performance in the results section („It should be highlighted the relationship between power output and cycling speed was linear outdoors and curvilinear indoors.“). Yes, if we plotted outdoors as a curve, we would obtain a different R2 value (it would be decreased) than the actual; however, we selected the linear regression due to its better fit. We did change the figure 4 to keep it simple (four different symbols - each one for each phase - would make it complicate).

Discussion

Line 25/26: environmental factors need to be mentioned in this sentence too.

Answer: We agree with the expert reviewer and added this aspect here („, and environmental factors such as wind and road’s surface [37,47].“).

Page 6 Line 5: please provide the 'so what' here. A smoother surface presents less rolling resistance than asphalt meaning the rider can attain higher speeds, for less power output. This needs to be stated. The rider may also have been able to adopt a more aerodynamic position indoors than outdoors, again a lower cross sectional area in a thermostatistically controlled environment reduces wind resistance in a consistent air pressure, also facilitating higher cycling speed.

Answer: We agree with the expert reviewer and added this aspect („A smoother surface decreases rolling resistance compared to asphalt, and consequently, a cyclist can attain faster speed for less power output. Moreover, a cyclist could adopt a more aerodynamic position indoors than outdoors with a lower cross-sectional area in a thermostatically controlled environment reducing wind resistance against a constant air pressure and, thus, increasing speed [47,48].“).

Line 6-10: I'm unsure what this adds by being placed here, it may fit one paragraph earlier, or may be superfluous.

Answer: We agree with the expert reviewer and placed it in the previous paragraph as suggested.

Line 14-15: you suggest a linear relationship to power/speed reduction but figure 4 shows a curve for the present dataset?

Answer: We agree with the expert reviewer and corrected it to „curvilinear“.

Line 16-19: there is a lot of information here, I'm unsure of all of its relevance to the present study unless it is written more directly and links to the present investigation e.g. Similarly, a non-linear decrease in power-output and cycling speed was seen in finishers of the RAAM, with a sharper decrease in cycling speed and power-output seen in the closing stages than the previous 20 checkpoints [5]. Similar pacing strategies have also been observed in mountain bike racing [47] and a 20-min uphill cycling time trial [48].

Answer: We agree with the expert reviewer and deleted this part.

Line 31-32: I don't think you can make this comment as you've not measured/reported energy intake, or gastrointestinal symptoms in any quantifiable way. The athlete was also fed a variety of drinks throughout the bout, so was this combination or a single drink the cause of the GI distress? I'd suggest removing this comment unless you have ample nutrition data that you're also willing to present in the present paper.

Answer: We agree with the expert reviewer and removed this sentence as suggested.

Line 37: What does monotonicity mean in this context? Is there an alternative definition?

Answer: We agree with the expert reviewer. Regarding https://bikeboard.at/Board/24-h-Bahn-Weltrekord-Christoph-Strasser-th237283 and the personal comment from the cyclist, we have no other expression.

Line 38: 'serious emotional distress' seems like it would cause cessation of exercise. Was this managed, measured or self-reported in some way? If so this needs to be explained in the methods and results. Recommend the ultra-running case study by Best et al., (2018) in Sports as a reference that assessed emotional factors over a 24hour duration, as opposed to the RAAM. 

Answer: We agree with the expert reviewer and added „Moreover, an incident of serious emotional distress after two hours of cycling was recorded by his support team.“ in the methods section, where a description of the race was made. In addition, we introduced the paper of Best et al. in the discussion and added that „Furthermore, it has been observed that perception of pain, freshness and motivation differed between a completed and a non-completed 100 mile running race [54].“.

Line 41-44: these sentences are in direct contradiction to each other. I suggest combining to 'caffeine use in cycling time trial performance has reported mixed effects [55-59], depending upon factors such as...'. If this is to be included, greater specificity is required.

Answer: We agree with the expert reviewer and revised this part as „Caffeine use in cycling time trial performance has reported mixed effects with some studies showing improved performance [55-57], whereas other research reported no improvement in time trial cycling following caffeine intake [58,59]. Such variation of the effectiveness of caffeinated drinks might depend upon factors such as heat, familiarization with caffeine consumption, quantity of caffeine consumption and exercise duration [55-59].“.

Lines 45-49. This paragraph needs to come earlier. You raise important points that would lead into your discussion around pacing strategies between cycling durations and disciplines.

Answer: We agree with the expert reviewer and moved it to a new fourth paragraph of the discussion.

I don't necessarily think you have a set of limitations as you did not design this experiment, it is a secondary analysis of the data produced by the athlete during the attempt. I would reframe this section as 'considerations'. This applies to things such as bike frame, testing surface, wheel choice, environment, nutritional strategies etc.

Answer: We agree with the expert reviewer and changed the “limitation” to “consideration” and added that the study “...relied on secondary analysis of data produced by the cyclist” in the paragraph before conclusions.

Conclusions

I think your conclusion emphasises the need for a direct comparison between trials as opposed to comparing correlation magnitudes and R2 values. A simple standardised difference in means at each phase of each attempt would suffice, this way you can attribute differences in global performance between attempts not just to environmental factors, but to performance in specific sections of the race.

Answer: We agree with the expert reviewer and added the recommended analysis.

Round 2

Reviewer 2 Report

I thank and commend the authors for the changes they have made in response to the feedback provided, and the timescale in which they have done so. I think the manuscript is much stronger and will be of great interest to cycling and endurance fans and researchers and am now happy to recommend accepting this in the present form. I look forward to seeing the published version.